# Three-Dimensional Characterization of Aortic Root Motion by Vascular Deformation Mapping

**DOI:** 10.3390/jcm12134471

**Published:** 2023-07-04

**Authors:** Taeouk Kim, Nic S. Tjahjadi, Xuehuan He, JA van Herwaarden, Himanshu J. Patel, Nicholas S. Burris, C. Alberto Figueroa

**Affiliations:** 1Department of Biomedical Engineering, University of Michigan, Ann Arbor, MI 48109, USA; taeouk@umich.edu; 2Department of Cardiac Surgery, University of Michigan, Ann Arbor, MI 48109, USA; nicasiut@med.umich.edu (N.S.T.); hjpatel@med.umich.edu (H.J.P.); 3Department of Radiology, University of Michigan, Ann Arbor, MI 48109, USA; xuhe@med.umich.edu; 4Department of Vascular Surgery, University Medical Center Utrecht, 3584 CX Utrecht, The Netherlands; j.a.vanherwaarden@umcutrecht.nl; 5Department of Surgery, University of Michigan, Ann Arbor, MI 48109, USA

**Keywords:** aortic root motion, VDM, 3D displacement, dynamic 3D CTA, age, LV/Ao angle

## Abstract

The aorta is in constant motion due to the combination of cyclic loading and unloading with its mechanical coupling to the contractile left ventricle (LV) myocardium. This aortic root motion has been proposed as a marker for aortic disease progression. Aortic root motion extraction techniques have been mostly based on 2D image analysis and have thus lacked a rigorous description of the different components of aortic root motion (e.g., axial versus in-plane). In this study, we utilized a novel technique termed vascular deformation mapping (VDM(D)) to extract 3D aortic root motion from dynamic computed tomography angiography images. Aortic root displacement (axial and in-plane), area ratio and distensibility, axial tilt, aortic rotation, and LV/Ao angles were extracted and compared for four different subject groups: non-aneurysmal, TAA, Marfan, and repair. The repair group showed smaller aortic root displacement, aortic rotation, and distensibility than the other groups. The repair group was also the only group that showed a larger relative in-plane displacement than relative axial displacement. The Marfan group showed the largest heterogeneity in aortic root displacement, distensibility, and age. The non-aneurysmal group showed a negative correlation between age and distensibility, consistent with previous studies. Our results revealed a strong positive correlation between LV/Ao angle and relative axial displacement and a strong negative correlation between LV/Ao angle and relative in-plane displacement. VDM(D)-derived 3D aortic root motion can be used in future studies to define improved boundary conditions for aortic wall stress analysis.

## 1. Introduction

Current guidelines for management of aortic diseases such as thoracic aortic aneurysms (TAA) and aortic dissection largely focus on maximal diameter as a primary metric of disease severity and risk [1,2]. Aortic diameter is an inherently static metric, typically performed in diastole [3]. However, the aorta is in constant motion due to the combination of cyclic loading and unloading with its mechanical coupling to the contractile left ventricle (LV) myocardium. Electrocardiogram (ECG)-gated computed tomography angiography (CTA) affords the opportunity to visualize motion of the thoracic aorta throughout the cardiac cycle and is capable of depicting such dynamic aortic root motion in a three-dimensional (3D) manner [4]. Abnormalities in aortic root motion have been proposed to be a potential risk factor for disease progression and complications in thoracic aortic disease [5,6,7,8].

Previous studies have characterized aortic root motion from dynamic imaging and have incorporated this data into methods to estimate wall stresses in the ascending thoracic aorta (ATAA). However, some studies have used idealized aortic models rather than patient-specific anatomy [5]. Furthermore, most techniques used to extract and quantify aortic root motion are based on 2D imaging and thus lack a comprehensive description of the multi-directional components of aortic root motion [5,6,7,8,9]. An accurate 3D assessment of cyclic motion of the ATAA could have significant implications for estimating mechanical properties of the aorta, informing computational simulations, endovascular device design, and, importantly, for refining diameter-based assessment of disease severity.

Vascular deformation mapping (VDM) is a validated medical image registration technique which allows for comprehensive assessment of the degree and extent of growth mapping of the aorta (VDM(G): VDM growth) [10,11,12] using longitudinal CTA data acquired at two different points during clinical surveillance. However, when applied to dynamic CTA data (i.e., time-resolved CTA), VDM allows for 3D assessment of the aortic deformation throughout the cardiac cycle (VDM(D): VDM dynamic).

The objective of this study is to utilize VDM(D) to accurately quantify aortic root motion in a 3D manner using patient-specific data in a cohort of patients with various manifestations of TAA disease (i.e., sporadic TAA and Marfan syndrome (MFS)) as well as non-aneurysmal controls. 

## 2. Materials and Methods

### 2.1. Study Population

All procedures were approved by the local institutional review board with a waiver of informed consent obtained for this retrospective study and were Health Insurance Portability and Accountability Act-compliant. On the basis of available high-quality, retrospectively electrocardiogram-gated CTA data, electronic medical records search software was used to identify patients aged 18 years and older with a clinical diagnosis of thoracic aortic aneurysm (TAA) that was either sporadic or associated with MFS. For comparison, we also identified a cohort of patients with available retrospective CTA data that had non-aneurysmal thoracic aortic dimensions (i.e., maximal diameter < 40 mm) as well as a group of patients who had undergone prior open surgical repair of their aortic root and/or ascending aorta. Exclusion criteria were non-ECG-gated acquisition, suboptimal thoracic aortic enhancement (<250 Hounsfield units), image slice thickness > 2.5 mm, or significant motion/respiratory artifacts preventing accurate delineation of the aortic wall and thus accurate registration in VDM analysis. At random, 51 patients meeting these criteria were included in this study and were divided into four groups: non-aneurysmal, sporadic TAA (henceforth referred to as “TAA”), MFS (henceforth referred to as “Marfan”), and ascending graft repair (henceforth referred to as “repair”). Patient demographics and clinical characteristics were collected through medical chart review. Maximum diameter measurements of the thoracic aorta were collected from clinical radiology reports. Peak systolic and end-diastolic phases of the ECG-gated CTA imaging were selected and extracted using OsirixMD [13]. The angle between the aortic annulus and the mitral annulus (LV/Ao angle) was also measured using OsirixMD for each patient. Figure 1 depicts the definition of LV/Ao angle adopted in this study, namely the angle (φ) between two lines perpendicular to the aortic and mitral annuli, respectively.

ECG-gated CTA examinations were performed on either 64- or 246-detector CTA scanners using helical (Lightspeed VCT or Discovery CT750HD; GE Healthcare) or axial (Revolution; GE Healthcare) acquisition modes. Dynamic CT imaging was acquired through the entire thoracic aorta, from lung apices to at least 2 cm below the celiac trunk.

### 2.2. VDM(D)

VDM employs b-spline deformable image registration techniques to quantify the 3D deformation of the aortic wall surface between two CTA images of a given subject. This approach has been previously applied to assess 3D aortic growth based on CTA images acquired at two time points spanning several years and has been validated in expert-rater and in silico phantom studies [10,11,12]. We refer to this growth assessment technique as VDM(G).

In this study, the VDM concept is expanded to study 3D aortic deformation between the diastolic and systolic phases extracted from the dynamic CTA images. Therefore, instead of assessing 3D growth over a long period of time, the method quantifies 3D deformation over the cardiac cycle. We refer to this technique as VDM(D). Figure 2 illustrates the VDM(D) workflow. First, the selected systolic and diastolic phase images are segmented using an in-house aortic segmentation neural network [14]. Next, rigid and deformable registrations were conducted to align the two segmentations. Registration accuracy was confirmed using a dual-channel plotting technique to assure alignment of the fixed diastolic and warped systolic configurations, as previously described [10]. Lastly, the 3D displacement field resulting from the deformable registration was used to perform a vertex-wise deformation of a triangulated mesh based on the diastolic configuration. The 3D displacement field between the diastolic and systolic aortic phases is amenable to performing engineering analysis of strains, stretches, and 3D motions of specific regions of the aorta, such as the aortic annulus.

### 2.3. Metrics Extracted from VDM(D)

#### 2.3.1. Defining a Suitable Location to Study Aortic Root Motion

VDM(D) provides the 3D displacement from diastole to systole over the entire surface of the thoracic aorta. However, for the purposes of this study, we extracted and analyzed aortic root motion at the sinuses of Valsalva in a plane perpendicular to the aortic centerline. Specifically, the analysis plane was placed at the level of the coronary arteries (i.e., coronary ostia) as this location provides a distinct anatomic feature for deformable registration during VDM(D) (Figure 3a). A normal vector to this analysis plane was obtained via cross-product of in-plane unit vectors. The green and magenta arrows in Figure 3b show normal vectors to the analysis plane in diastole and systole, respectively.

#### 2.3.2. Displacement

Figure 3a represents the total, axial, and in-plane displacement vectors in 3D space. The total displacement (black arrow) was obtained by averaging the 3D displacement field as given by VDM(D) over the entire analysis plane. The axial displacement (orange arrow) was defined as the projection of the total displacement vector in the direction of the diastolic normal vector (green arrow in panel b). Conversely, the in-plane displacement (purple arrow) was defined as the projection of the total displacement in the direction perpendicular to the diastolic normal vector. In addition, relative displacements in axial and in-plane directions can be defined as follows:(1)$$Relative\;axial\left(in\text{-}plane\right)dislacement=\frac{axial\left(in\text{-}plane\right)displacement\;[\mathrm{mm}]}{total\;displacement\;[\mathrm{mm}]}$$

#### 2.3.3. Distensibility and Area Ratio

Distensibility is a metric that reflects aortic stiffness as it includes changes in both strain and pressure. Here, the ratio between diastolic and systolic analysis plane areas was obtained to calculate the distensibility (Figure 3c) using the following formula [15]:(2)$$Distensibility\left[10^{-3}\;\mathrm{mmH}g^{-1}\right]=\frac{A_{sys}-A_{dia}}{A_{dia}}\times \frac{1}{PP}=(area\;ratio-1)\times \frac{1}{PP}$$
where $A_{sys}$ and $A_{dia}$ refer to the systolic and diastolic annulus areas, respectively, and PP is the pulse pressure. Area ratio is $A_{sys}/A_{dia}$.

#### 2.3.4. Axial Tilt and Aortic Rotation

The aortic root experiences a complex motion involving rotation and twisting around multiple axes due to its mechanical coupling with the contracting left ventricle. To characterize these complex rotations, in this paper we defined the following metrics.

Axial tilt α, defined as the angle between the diastolic (green arrow) and systolic (magenta arrow) normal vectors and their corresponding analysis plane areas (Figure 3b).

Aortic rotation θ, defined as the angle between two vectors from the centroid of the diastolic and systolic analysis plane areas and a reference point such as the initial point of the centerline of the left coronary artery (Figure 3d). Here, the green arrow represents the reference diastolic vector, and the magenta arrow represents the projection of the reference systolic vector onto the diastolic annulus plane. This aortic rotation θ represents the cyclic twisting motion of the aorta.

### 2.4. Statistical Analysis

Due to the small sample size of the different groups considered in this study (~10 subjects per group), we could not assume normal distribution behavior for each group. Therefore, a Kruskal–Wallis test was performed to compare different groups [16]. A *p*-value < 0.05 was considered significant. Correlations between variables were assessed using linear regression. Again, a *p*-value < 0.05 was considered a strong correlation. The ‘kruskalwallis’ and ‘fitlm’ MATLAB functions were used for the Kruskal–Wallis test and linear regression, respectively.

## 3. Results

### 3.1. Patient Demographics

The 51 subjects included in this study were divided into four groups, as follows: 13—non-aneurysmal; 15—TAA; 11—Marfan; and 12—repair. Patient demographics are shown in Table 1. Overall mean age was 55 ± 14.4 years old. Age distribution among the four groups is depicted in Figure 4 and Table 1. Age was significantly different between groups except between Marfan and non-aneurysmal groups, with the repair group demonstrating the highest mean age (Figure 4). Mean systolic/diastolic and pulse pressures were 127/71 mmHg and 56 mmHg, respectively, and did not differ across groups.

TAA and Marfan groups demonstrated larger maximal sinus compared to the non-aneurysmal and repair groups (*p*-value < 0.001). Maximal diameter at the sinotubular junction (STJ) and mid-ascending aorta (MAA) was higher in the TAA group compared to the other groups (*p*-value < 0.001), concordant with the predilection for MAA dilation in sporadic TAA. LV/Ao angle was lowest in the repair group.

### 3.2. Aortic Root Motion Metrics

Figure 5 shows diastolic and systolic aortic geometries extracted from VDM(D) as well as the corresponding sinus contours at the analysis plane for representative subjects in each group (e.g., subject with motion patterns closest to the mean of the group). The 3D geometries and sinus contours demonstrate downward (axial) motion and contour expansion in systole across all groups and subjects. In addition, in-plane motion was apparent for all subjects but was lowest in the repair group. The non-aneurysmal and repair subjects show the largest and smallest axial displacements, respectively. The largest component of the aortic root motion in the repair subjects was in the in-plane direction.

Table 2 and Figure 6 summarize the results for the aortic root motion metrics across all patients and groups. On average, the repair group showed approximately 3.5 mm less total displacement than the other three groups; however, there was no significant difference between non-aneurysmal, TAA, and Marfan groups. The Marfan group showed the largest heterogeneity in total displacement (range: 1.77 to 10.03 mm). In contrast, the repair group demonstrated the narrowest range of total displacements (range: 1.93 to 4.99 mm, Figure 6a).

The axial and in-plane displacements show similar trends to the total displacement. The repair group showed significantly smaller axial and in-plane displacement than the other groups (Figure 6b,c). Table 2 also summarizes the relative axial and in-plane displacements (i.e., the directional component normalized by the total displacement). The repair group showed smaller relative axial displacement and larger relative in-plane displacement than the other three groups (*p*-value < 0.01). The non-aneurysmal group was the only group showing larger relative axial displacement (0.72) compared to relative in-plane displacement (0.66).

There was no statistical difference in axial tilt between the four groups (Figure 6d). The repair group showed smaller aortic rotation values than the other groups (Figure 6e), but there was no statistical difference in rotation between non-aneurysmal, TAA, and Marfan groups. Figure 6f represents the distensibility results. The non-aneurysmal group showed higher median distensibility compared to the Marfan and repair groups; however, there was no statistical difference between non-aneurysmal and TAA groups. The TAA group displayed higher median distensibility compared to the repair group. The Marfan group showed the largest heterogeneity, with distensibility values ranging from 0.2 to 2.5 × 10^−3^ mmHg^−1^. In contrast, the repair group shows the smallest heterogeneity in distensibility, ranging from 0.4 to 1.3 × 10^−3^ mmHg^−1^.

### 3.3. Correlation with Age

Figure 7 summarizes the correlation between age and total displacement (panel a) and distensibility (panel b) for each group. Although each group showed a negative correlation between displacement and age, these correlations were weak–moderate and not statistically significant (Figure 7a). In contrast, there was a strong negative correlation between distensibility and age in the non-aneurysmal group but not in the other groups. In a subgroup of young patients (<40 years), distensibility was lower in Marfan compared to non-aneurysmal patients (1.73 vs. 4.04, *p*-value = 0.025).

### 3.4. LV/Ao Angle Results

The LV/Ao angle (φ) represents the alignment between the aortic annulus and long-axis of the left ventricle (see Figure 1). The smaller this angle, the more perpendicularly oriented the heart and the aorta. Figure 8a summarizes the distribution of LV/Ao angles for each group. The repair group showed an approximately 5-degree smaller LV/Ao angle than the other groups. There was no statistical difference between non-aneurysmal, TAA, and Marfan groups. Combining the results in Figure 4 and Figure 8a, we can appreciate that, in general, smaller LV/Ao angles are associated with older age.

Figure 8b,c illustrates the correlation between the LV/Ao angle and relative axial and in-plane displacements, respectively, for all subjects of all groups. Panel b shows that there was a strong positive correlation between LV/Ao angle and relative axial displacement. Conversely, Panel c shows a moderate negative correlation between LV/Ao angle and relative in-plane displacement.

## 4. Discussion

*High-level summary*: In this study, 3D aortic root motion was characterized using dynamic CTA data and VDM(D) analysis. Aortic root motion metrics such as axial and in-plane displacements, area ratio/distensibility, axial tilt, aortic rotation, and LV/Ao angles were extracted and compared for four different subject groups: non-aneurysmal, TAA, Marfan, and repair. Our results revealed differences in aortic root motion metrics, most notably between the repair group and other groups with native ascending aortas. The repair group showed the smallest aortic root displacement, aortic rotation, and distensibility compared to other groups. There was no difference in axial tilt between groups. The repair group was also the only group that showed a larger relative in-plane displacement than relative axial displacement, likely explained by the lowest (i.e., more perpendicular) LV/Ao angle in this group. The Marfan group showed the largest heterogeneity in aortic root displacement, distensibility, and age, compatible with well-reported heterogeneity in this syndrome. We also showed that there were moderate–strong correlations between LV/Ao angle and relative axial and in-plane displacements.

This study adds several clinically relevant advances and insights. First, our proposed method is performed on ECG-gated dynamic CTA data acquired as part of routine clinical care. Thus, this is a technique that can be applied to routine clinical imaging without the need for non-standard or research imaging techniques. Second, given that stresses in the ascending aorta are in part determined by the downward pulling on the aortic root by the LV and that there is a large body of evidence connecting aortic stresses to disease progression and development of complications such as aortic dissection, the analytic tools to assess root motion may yield important insights into a patient’s disease severity that may have implications for risk stratification. Additionally, our methods could be employed to better understand the biomechanical effects of stiff fabric or metallic endografts on aortic function. Clearly, making these inferences will require substantial additional research, but the immediate applicability of techniques to clinical data will greatly lessen barriers to these translational studies.

*Repair group*: Table 2 and Figure 6 show that the repair group presented smaller and more homogeneous displacements and rotations than the other groups. These findings are consistent with previous studies [17], indicative of the higher graft stiffness relative to the native aortic tissue [18,19]. Distensibility was also lowest in the repair group, an expected finding given the very stiff properties of synthetic fabric vascular grafts [20,21,22].

*Marfan group*: The Marfan group had the largest heterogeneity in displacement and distensibility (Figure 6) as well as in age (Figure 4). Such heterogeneity has been previously documented among patients with MFS [23]. While a pathogenic mutation in the fibrillin-1 gene is pathognomonic of MFS disease, it is well recognized that large degrees of phenotypic heterogeneity exist between patients, an observation that aligns with the variability we observed in root motion and mechanical metrics studied in this paper [24]. From a clinical perspective, methods to define the phenotypic severity of disease are greatly desirable, particularly at earlier stages of disease, to allow for improved estimates of risk and better-informed decisions surrounding prophylactic aortic repair. Given that aortic aneurysm is one of the defining manifestations of MFS disease and a leading cause of morbidity and mortality, novel methods that can better understand the severity of aortic disease may be impactful. A prior study using echocardiography to assess root distensibility in MFS demonstrated the same heterogeneity in these metrics that we observed [25]. However, this study did not examine the additional root motion metrics we considered in our work. Thus, while there is still much to learn about the clinical significance of such findings, we are encouraged by both replicating previous observations and identifying differences in root motion and distensibility metrics in a disease that is known to present highly variable manifestations between different affected individuals.

*Distensibility*: Our results revealed that the Marfan and repair groups had lower aortic root distensibility than the non-aneurysmal and TAA groups, in alignment with prior reports [21,26]. Interestingly, we observed no statistical difference between non-aneurysmal and TAA groups (Figure 6f), in contrast with prior studies that have reported higher aortic stiffness in TAA compared to non-aneurysmal aortas [27,28,29]. This discrepancy may be explained by several factors. First, the location where aortic motion is extracted in our study is at sinuses of Valsalva, which is different from the location of maximum aneurysm diameter in our TAA subjects, suggesting a potential gradient in distensibility (and therefore stiffness) from the aortic root to the aneurysm location in TAA subjects. Secondly, we may have simply failed to capture a statistical difference in these groups owing to relatively small group sizes and the confounding effects of age on aortic stiffness in the non-aneurysmal group.

*LV/Ao angle*: Previous studies have reported a negative correlation between age and LV/Ao angle [30,31]. Our results are consistent with these studies (see Appendix A, Figure A1). To better understand the correlation between LV/Ao angle and root motion metrics, we examined relative aortic displacements normalized to total displacement given that LV/Ao angle would be expected to affect the directional proportion of root motion more than its absolute degree (more reflective of the underlying aortic pathology). Further, normalizing the displacements allowed us to mitigate the relationship between age and aortic displacement and therefore to directly compare all subjects across groups. 

*Variation of metrics with age*: Previous studies have reported clearly demonstrated increasing aortic stiffness with aging [32,33,34]. Therefore, the negative correlation between age and distensibility (inversely related to stiffness) in the non-aneurysmal group (Figure 7b) is consistent with prior results. However, the TAA, Marfan, and repair groups did not show significant correlations between distensibility and age. All three groups presented lower distensibility compared to the non-aneurysmal group. For instance, young (<40 years) Marfan subjects had lower distensibility compared to non-aneurysmal patients (1.73 vs. 4.04, *p*-value = 0.025). This suggests that the effects of aortic stiffening related to these patients’ aneurysmal disease supersedes the effects of age-related aortic stiffening. 

Figure 7a shows that there was no strong correlation between age and aortic root displacement. Only the TAA group suggested a pattern of larger total displacements for younger subjects. However, this TAA group is highly heterogenous with six bicuspid aortic valve, five aortic stenosis, and three aortic insufficiency patients. Unlike distensibility, which is dictated by aortic properties only, aortic root displacement results from the interactions between heart and aorta. Previous studies have reported that there is no significant correlation between age and LV ejection fraction [35,36]. Therefore, one could argue that aortic root motion is independent of age unless heart function is compromised.

*Correlation with aortic diameter*: Aortic diameter has been widely used as a clinical metric to assess aortic disease severity and progression [25,37,38]. However, numerous studies have demonstrated its poor predictive value [39,40,41]. Our results agree with such studies as we have observed no significant correlation between sinus diameter, distensibility, and aortic displacement (see Appendix A, Figure A2).

### 4.1. Displacement Extraction Comparison between 2D and 3D

Most studies thus far have characterized aortic root motion using 2D approaches [5,7,8,9,17,42,43]. However, extracting aortic root motion from 2D images without accounting for the full 3D motion of the aorta may provide a distorted assessment. Figure 9 illustrates diastolic (blue) and systolic (red) aortic geometries extracted from VDM(D) for a non-aneurysmal subject. The motion is observed under two different views. View 1 shows that there is an axial (downward) motion and almost no in-plane motion. However, view 2 shows that there are both axial and in-plane displacements. This is a simple demonstration of how a VDM(D) analysis of dynamic CTA data can more fully capture 3D motion of the root compared to 2D approaches.

### 4.2. Implications for Aortic Wall Stress Analysis

Three-dimensional aortic root motion extracted from VDM(D) can be used to provide improved boundary conditions for aortic wall stress analysis [44,45]. Most previous computational studies of ascending aortic stress hold the inlet boundary as fixed (no root motion) [46,47,48,49,50]. Few studies have investigated the effect of aortic root motion on ascending aortic wall stress under simplified and unidirectional (e.g., axial) aortic root motion assumptions [5,7,51]. A recent study demonstrated that in-plane displacements contribute significantly to the stress level in the ascending aorta [8]. However, this study considered simplified, circumferentially uniform values of in-plane displacements extracted from 2D dimensional data. In reality, the aortic root has circumferentially variable in-plane and axial displacements which can be fully quantified by VDM(D). This 3D information could be used to define improved boundary conditions for aortic wall stress analysis.

### 4.3. Limitations

The small number of subjects per group is one limitation of this study which may result in a lack of strong correlations for some of the reported aortic root metrics in some of the subject groups. The composition of the TAA group was heterogenous as one third of the patients had BAV, one third had aortic stenosis, and one fifth had aortic insufficiency. These comorbidities may have an impact on aortic root motion. However, the primary objective of this study was to assess feasibility of VDM(D) rather than provide a comprehensive assessment of metrics across different patient groups and characteristics. Future studies will expand the sample size of the TAA group and thus enable us to define subgroups according to the presence or absence of valvular disease.

Additionally, the different patient groups had statistically significant differences in age. As expected, the TAA and repair groups were older than non-aneurysmal and Marfan groups (Figure 4). These trends therefore reflect the strong association between sporadic/degenerative TAA and age [52]. The Marfan group was youngest overall and had the largest variability in age (from mid-20s to nearly 60 years old), consistent with the well-known variability in phenotypes among Marfan patients [20]. Future studies using groups matched on age and other characteristics will be important to more fully understand the unique contribution of such factors on the described aortic root motion metrics.

Lastly, this study focused on establishing the feasibility of VDM(D) analysis using dynamic CTA data rather than comparing our method against other 2D root motion methods using cardiac MRI (CMR) or echocardiography. Such a comparative study, while interesting, would require prospective studies since it is highly unlikely that a patient would have analyzable images from all three modalities to allow assessment of inter-modality agreement.

## 5. Conclusions

VDM(D) analysis of dynamic CTA data enables a rigorous characterization of 3D aortic root motion. Directional aortic displacements, area ratio and distensibility, axial tilt, and aortic rotation can be extracted across a variety of TAA etiologies as well as in post-repair groups. Our results revealed differences in aortic root motion metrics, most notably between the repair group and other groups with native aortic tissue. The non-aneurysmal group showed a negative correlation between age and distensibility, consistent with widely reported age-related aortic stiffening. Additionally, our results demonstrated that the LV/Ao angle is an important determinant of the proportions of axial and in-plane displacements and should thus be included in future studies focused on dynamic assessment of the ascending aorta. 

## Figures and Tables

**Figure 1 jcm-12-04471-f001:**
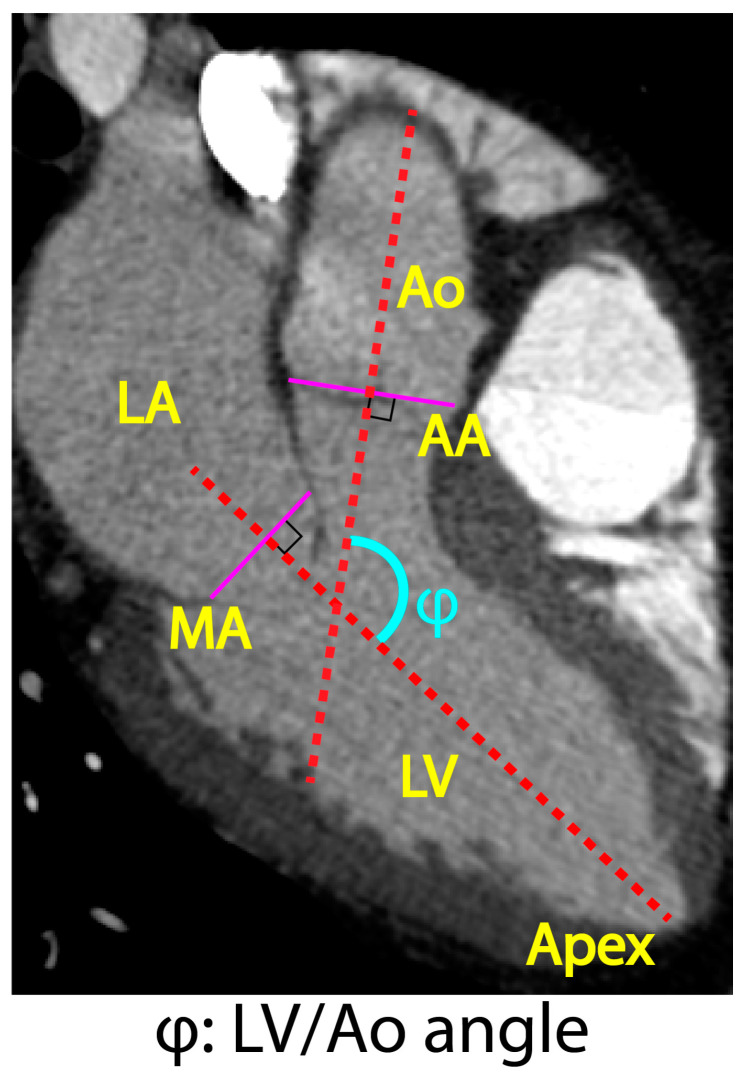
Definition of LV/Ao angle (φ) adopted in this study. LV: left ventricle, MA: mitral annulus, LA: left atrium, AA: aortic annulus, Ao: aorta. Purple lines indicate MA and AA. The LV/Ao angle (φ) is the angle between the lines perpendicular to the aortic and mitral annuli, respectively.

**Figure 2 jcm-12-04471-f002:**
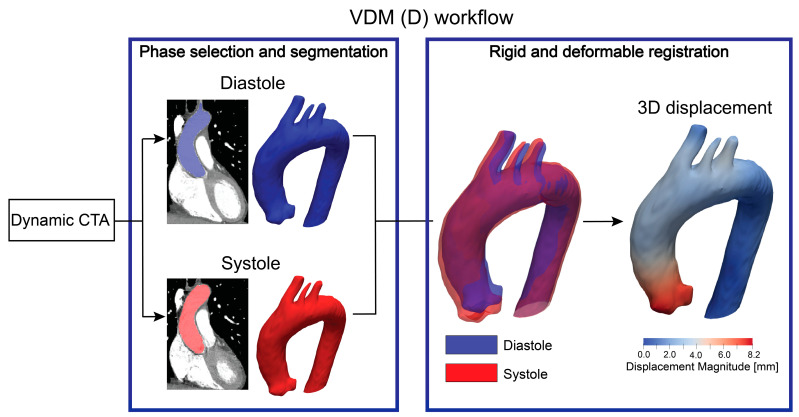
VDM(D) workflow.

**Figure 3 jcm-12-04471-f003:**
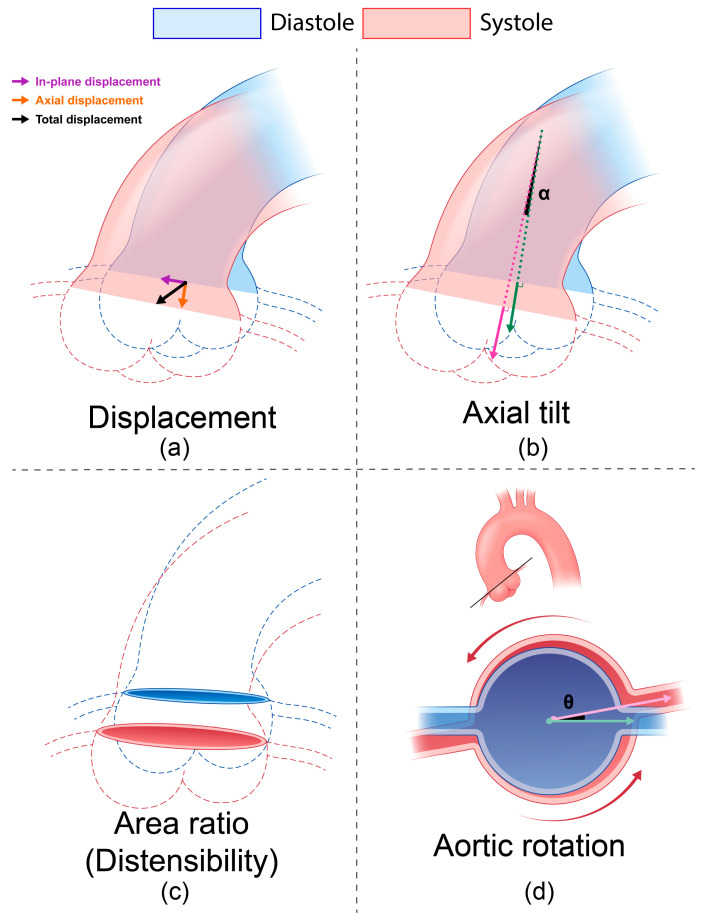
Four metrics of aortic root motion considered in this study, all calculated from 3D displacement extracted from VDM(D): (**a**) total, axial, and in-plane displacements, (**b**) axial tilt, (**c**) area ratio (distensibility), and (**d**) aortic rotation.

**Figure 4 jcm-12-04471-f004:**
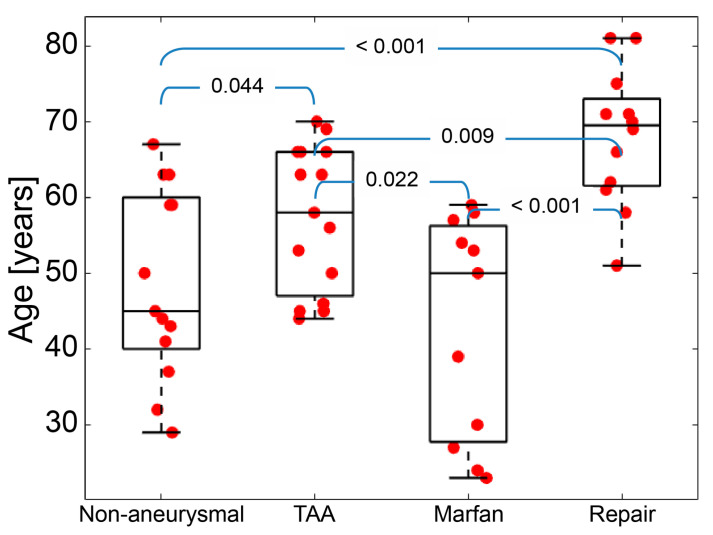
Age whiskers box plots for different groups; *p*-values are given at the top of the blue brackets.

**Figure 5 jcm-12-04471-f005:**
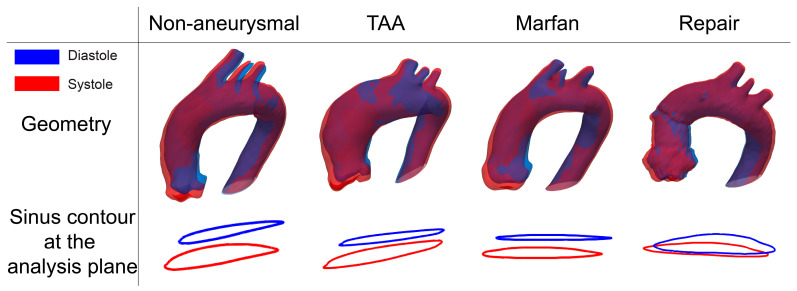
Diastolic (blue) and systolic (red) aortic geometries extracted from VDM(D) and the corresponding sinus contours at the analysis plane for representative subjects in each group.

**Figure 6 jcm-12-04471-f006:**
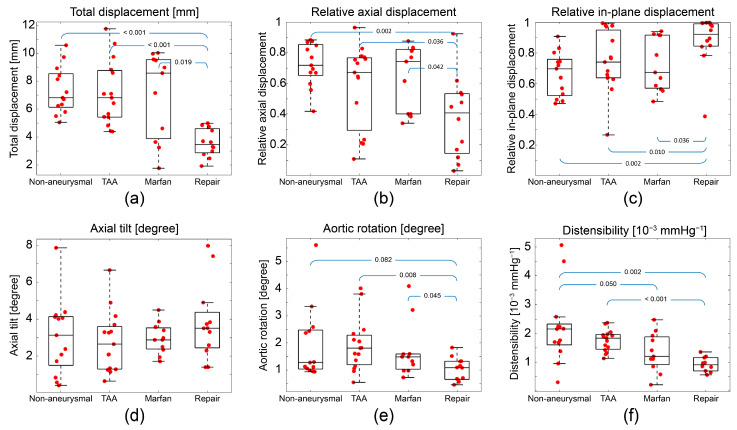
Whiskers box plots for the different subject groups: (**a**) total displacement, (**b**) axial displacement, (**c**) in-plane displacement, (**d**) axial tilt, (**e**) aortic rotation, and (**f**) distensibility; *p*-values are given at the top of the blue brackets.

**Figure 7 jcm-12-04471-f007:**
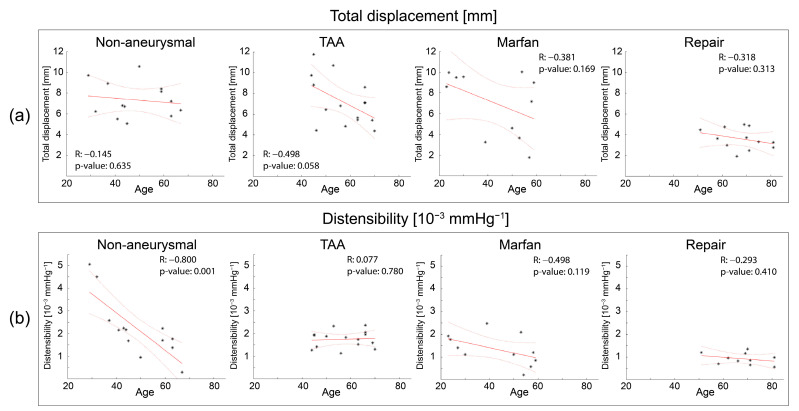
Scatter plots correlating (**a**) total displacement vs. age and (**b**) distensibility vs. age for the different subject groups. Solid and dotted red lines are fit and confidence bounds, respectively.

**Figure 8 jcm-12-04471-f008:**
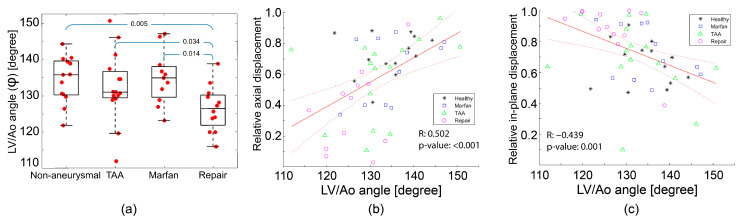
LV/Ao angle results. (**a**) Whiskers box plots of LV/Ao angle for each group; *p*-values are given at the top of the blue brackets. (**b**) Scatter plot correlating relative axial displacement and LV/Ao angle. (**c**) Scatter plot correlating relative in-plane displacement and LV/Ao angle. Solid and dotted red lines are fit and confidence bounds, respectively.

**Figure 9 jcm-12-04471-f009:**
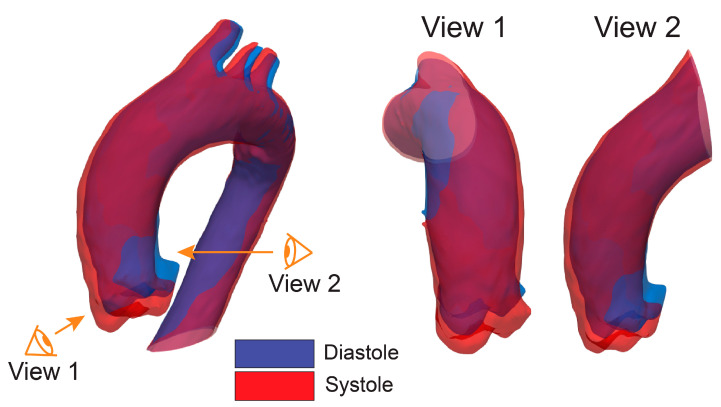
Diastolic (blue) and systolic (red) aortic geometries extracted from VDM(D) for a non-aneurysmal subject. Two different 2D views provide different characterizations of aortic root motion.

**Table 1 jcm-12-04471-t001:** Patient characteristics.

Characteristic (*n* = 45)	Non-Aneurysmal (*n* = 13)	TAA (*n* = 15)	Marfan (*n* = 11)	Repair (*n* = 12)	*p*-Value
Age (years)	48.6 ± 12.5	57.3 ± 9.5	43.1 ± 14.7	68.0 ± 9.0	<0.01
Female (n)	6	5	6	2	-
BP (systolic) (mmHg)	130 ± 22	125 ± 19	125 ± 17	129 ± 11	0.86
BP (diastolic) (mmHg)	77 ± 15	68 ± 8	69 ± 10	71 ± 10	0.21
Pulse pressure (mmHg)	53 ± 12	56 ± 19	56 ± 14	58 ± 16	0.86
HTN (*n*)	8	8	7	6	-
BAV (*n*)	0	6	1	0	-
AS (*n*)	0	5	0	0	-
AI (*n*)	0	3	0	0	-
CAD (*n*)	2	2	0	3	-
Hyperlipidemia (*n*)	2	7	2	5	-
Diameter (sinus) (mm)	32 ± 4	43 ± 5	43 ± 6	36 ± 5	<0.01
Diameter (STJ) (mm)	28 ± 3	43 ± 5	36 ± 6	32 ± 3	<0.01
Diameter (MAA) (mm)	30 ± 4	45 ± 4	33 ± 4	32 ± 3	<0.01
LV/Ao angle (degrees)	132.5 ± 9.5	134.4 ± 6.4	134.9 ± 7.4	126.4 ± 6.4	<0.01

Mean ± standard deviation. TAA = thoracic aortic aneurysm, BP = blood pressure, HTN = hypertension, BAV = bicuspid aortic valve, AS = aortic stenosis, AI = aortic insufficiency, CAD = coronary artery disease, STJ = sinotubular junction, MAA = mid-ascending aorta, LV/Ao angle = left-ventricular/aortic root angle.

**Table 2 jcm-12-04471-t002:** Aortic root metrics for different subject groups.

	Non-Aneurysmal	TAA	Marfan	Repair
Sinus contour at the analysis plane (blue= diastolic, red= systolic)	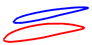	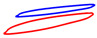	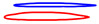	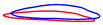
Displacement[mm]	Total	7.34 ± 1.69	7.14 ± 2.30	7.01 ± 3.09	3.60 ± 1.00
Axial	5.20 ± 1.17	4.23 ± 2.27	4.84 ± 2.88	1.37 ± 1.14
In-plane	4.98 ± 1.93	5.33 ± 2.32	4.79 ± 2.08	3.13 ± 1.06
Relative axial displacement	0.72 ± 0.14	0.58 ± 0.27	0.63 ± 0.21	0.38 ± 0.26
Relative in-plane displacement	0.66 ± 0.14	0.68 ± 0.25	0.73 ± 0.17	0.88 ± 0.17
Axial tilt (degree)	3.06 ± 2.06	2.79 ± 1.65	2.96 ± 0.83	3.83 ± 2.08
Aortic rotation (degree)	1.93 ± 1.35	1.92 ± 0.97	1.70 ± 1.02	1.03 ± 0.43
Distensibility (10^−3^ mmHg^−1^)	2.21 ±1.30	1.75 ± 0.37	1.34 ± 0.68	0.93 ± 0.26

Mean ± standard deviation. TAA = thoracic aortic aneurysm. Relative axial (in-plane) displacement = axial (in-plane) displacement/total displacement.

## Data Availability

The data presented in this study are available on request from the corresponding author.

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
