# Peer review of "Three-Dimensional Characterization of Aortic Root Motion by Vascular Deformation Mapping"

_jcm, 2023, doi:10.3390/jcm12134471_

Round 1
Reviewer 1 Report
The authors utilized a novel technique termed vascular deformation mapping VDM(D) to extract 3D aortic root motion from dynamic computed tomography angiography images. Aortic root displacement (axial and in-plane), area ratio and distensibility, axial tilt, aortic rotation, and LV/Ao angles were extracted and compared for four different subject groups: non-aneurysmal, TAA, Marfan, and repair. They revealed a positive correlation between LV/Ao angle and relative axial displacement and a strong negative correlation between LV/Ao angle and relative in-plane displacement. VDM(D)-derived 3D aortic root motion can be used in future studies to define improved boundary conditions for aortic wall stress analysis.
Here are two mirror questions:
(1) Will this VDM(D) work for aortic dissection?
(2) Figure 4 seems to be unnecessary in this manuscript.
Reviewer 2 Report
The purpose of this study is to evaluate the three-dimensional characterization of aortic root by vascular deformation mapping, which is a promising technique. However, some issues may limit this work.
1. CTA based vascular deformation mapping can theoretically provide a more comprehensive evaluation of aortic root motion compared to two-dimensional imaging. However, unfortunately, this article seems to focus on the comparison of aortic motion between different patients using vascular deformation mapping, without comparing this new technology with traditional two-dimensional imaging.
2. Although this new technology provides a more comprehensive assessment of aortic root movement, it is also more time-consuming, expensive, and radiative compared to traditional ultrasound. The authors need to supplement the potential population for clinical adaptation of this technology.
Minor editing of English language required
Reviewer 3 Report
The article presents an interesting concept and is overall well organized and the quality of presentation is high. Materials and Methods are well explained and clear. The statistics are good, the quality of the graphics is outstanding. The sample size is small for a pilot study, therefore the observation at lines 169-173 is appropriate.
I found very interesting the correlation with age at lines 240-247, I wonder if there is any clinical implication for this finding.
The statement at lines 294-295 should be explained in depth.
Another question I would like to ask to the authors is: what is the clinical implication for the finding explained at lines 327-329?
In conclusion, the study is well organized and the quality of presentation is high. Some limitations are that the sample size is small and the study is retrospective; it would be interesting to try to implement randomization for future studies.
Reviewer 4 Report
Dear authors,
Congratulations on your article “Three-Dimensional Characterization of Aortic Root Motion by Vascular Deformation Mapping”.
The Discussion and Conclusion chapters should be reduced in terms of length, considering how dense they are.
Additionally, a paragraph should be added in the beginning of the Discussion summarizing the results. More information should also be included empathizing what this papers adds and its positive features and contributions in terms of clinical relevance.
The sentences from line 63 to line 70 should be removed.
Best Regards.
